# Low Vitamin D Status Relates to the Poor Response of Peripheral Pulse Wave Velocity Following Acute Maximal Exercise in Healthy Young Men

**DOI:** 10.3390/nu14153074

**Published:** 2022-07-26

**Authors:** Liang-You Chen, Chun-Wei Wang, Lu-An Chen, Shih-Hua Fang, Soun-Cheng Wang, Cheng-Shiun He

**Affiliations:** 1Department of Athletic Sports, National Chung Cheng University, Minxiong 621301, Taiwan; liangyou88@alum.ccu.edu.tw (L.-Y.C.); yoyoyoo0714@gmail.com (C.-W.W.); 1003054@ntsu.edu.tw (L.-A.C.); grcscw@ccu.edu.tw (S.-C.W.); 2Department of Sports Performance, National Taiwan University of Sport, Taichung City 404401, Taiwan; shfang@ntus.edu.tw

**Keywords:** arterial stiffness, 25(OH)D, vascular function

## Abstract

The primary objective of this study was to determine the effects of vitamin D levels on peripheral pulse wave velocity (pPWV) following acute maximal exercise in healthy young adults. Fifty male healthy adults from National Chung Cheng University participated in the study. Participants were divided into the 25-hydroxyvitamin D (25(OH)D) sufficiency group (*n* = 28, 25(OH)D ≥ 50 nmol/L) and deficiency group (*n* = 22, 25(OH)D < 50 nmol/L). The acute maximal exercise was performed using an incremental cycling test to exhaustion. Additionally, the pPWV and blood pressure were obtained at rest and 0, 15, 30, 45, 60 min after acute maximal exercise. The results show that 25(OH)D deficiency group had higher pPWV at post-exercise (5.34 ± 0.71 vs. 4.79 ± 0.81 m/s, *p* < 0.05), post-exercise 15 min (5.13 ± 0.53 vs. 4.48 ± 0.66 m/s, *p* < 0.05) and post-exercise 30 min (5.26 ± 0.84 vs. 4.78 ± 0.50 m/s, *p* < 0.05) than the sufficiency group. Furthermore, there was a significant inverse correlation between 25(OH)D levels and pPWV following acute maximal exercise. Our study demonstrated that low vitamin D status relates to the poor response of pPWV following maximal exercise in healthy young men. Vitamin D deficiency may increase the risk of incident cardiovascular events after acute exhaustive exercise, even in healthy and active adults.

## 1. Introduction

Vitamin D deficiency is reported to be a global problem caused by limited exposure to natural sunlight [1,2]. It is a highly prevalent condition affecting about 50% of the U.S. population that is associated with hypertension, insulin resistance, and left ventricular hypertrophy [3]. Additionally, a report by the UK Scientific Advisory Committee on Nutrition shows that 22–24% of individuals aged 19–64 years in the UK are vitamin D deficient [4]. The Institute of Medicine (IOM) and the Endocrine Society defined a cutoff level of 50 nmol/L as 25-hydroxyvitamin D (25(OH)D) deficiency. They considered bone health and dietary reference intakes and found that 50 nmol/L is the requirement for vitamin D in nearly all normal healthy individuals [5]. Previous clinical studies have indicated that vitamin D deficiency could impair vascular function and vascular compliance, as evidenced by increased arterial stiffness [1,6]. In addition, the study has illustrated that individuals with vitamin D levels below 50 nmol/L have an increased risk of cardiovascular diseases (CVD) and all-cause mortality in the general population [7].

Increased pulse wave velocity (PWV) is the index of arterial stiffness and the major risk factor for CVD. PWV is considered to be a simple, reliable and validated measure of arterial stiffness [8,9]. A higher PWV results in lower vascular assignability and compliance; therefore, it corresponds to higher arterial stiffness [10]. Measurements of arterial stiffness were divided into central and peripheral PWV. Central PWV is usually measured by near-cardiac arteries, such as the carotid–femoral pulse wave velocity (cf-PWV); peripheral PWV (pPWV) is measured by a body limb pulse wave, such as finger–toe pulse wave velocity (ft-PWV). Previous studies have shown that the pPWV measurement is based on peripheral arteries signals and highly correlated with methods for assessing central arteries stiffness [8]. Moreover, the study suggests that the agreement between the pPWV values and the gold-standard method is outstanding, and pPWV was significantly associated with CVD [11].

The acute maximal exercise could cause significant changes in arterial stiffness and may reveal vascular abnormalities not apparent at rest [12,13]. Naka et al. [14] indicated that the stiffer intrinsic wall properties of arteries due to collagens and proteoglycans accumulation may only be detectable in the vasodilated state, such as after high intensity exercise. Acute maximal exercise has been shown to temporarily increase central arterial stiffness after exercise in young normal-weight adults [13,15]. Several studies in healthy young cohorts have shown that pPWV immediately and significantly decreases in post-exercise, which persists during the period of recovery for approximately 30 min, while central PWV remains unchanged [16]. Thus, previous studies revealed that the use of pPWV may reveal more obvious alterations in vascular function during post-exercise recovery. On the other hand, the study suggested that estrogen of the female menstrual cycle affects arterial stiffness and vascular response post-exercise. Accordingly, we did not recruit females as participants in our study [17].

Vitamin D deficiency is a risk factor for endothelial dysfunction and an early event in the development of atherosclerosis [18]. Additionally, vitamin D deficiency is associated with increased PWV in terms of arterial stiffness [19]. Whether vitamin D deficiency is associated with changes in pPWV after acute maximal exercise is currently unclear. Therefore, the primary objective of this study was to determine the effects of vitamin D status on pPWV and the relationship between vitamin D levels and pPWV following acute maximal exercise in healthy adults.

## 2. Materials and Methods

### 2.1. Participants

A priori power analysis with G*Power (version 3.1.9.7, Heinrich Heine University, Düsseldorf, Germany) was performed, using a predefined power of 0.80, an effect size of 0.94 and an alpha level of 0.05 [20]. According to the parameters, this led to a required sample size of at least 15 participants in each group.

Fifty healthy young males aged 21.14 ± 1.95 years regularly participated in exercise training in National Chung Cheng University, Taiwan (latitude 23.5° N). All subjects volunteered in this study, conducted from October to December. The sunlight in Southeast Asia during this period would not significantly alter the participants’ vitamin D status [21]. They needed to provide their self-reported training loads, which averaged 527.80 ± 284.75 min/week (mean ± SD), and were surveyed by pre-screening questionnaires. The health screening questionnaire was written by participants before testing. They were also required not to take any nutritional supplements or regular medication for 3 months prior to the study. All participants were fully informed of the study precautions and all experimental procedures. Participants provided written consent to participate in the study, which was earlier approved by the National Chung Cheng University ethical advisory committee in accordance with the 1964 Helsinki Declaration and its later amendments or comparable ethical standards (No: CCUREC109040701). Participants had to be free of health problems or infections within the past 2 weeks, engage in regular exercise (at least 3 sessions per week and at least 3 h of total moderate/high intensity training time), and be between the ages of 18 and 35. Participants were excluded if they smoked or had one of the following types of diseases: renal, neurological cardiac, hepatic, pulmonary, or gastrointestinal. In addition, participants were not permitted to take any other supplements.

### 2.2. Procedures

The information from the study was received by the participants before the first laboratory visit. The participants arrived after a 3 h fast from 09:00 am–12:00 pm, and their weight and height were recorded. Participants were required to abstain from any strenuous physical activity for 24 h before coming to the laboratory. Participants completed the health-screening questionnaire as an exclusion and inclusion criterion prior to signing the informed consent form. Subsequently, a quiet venous blood sample (6 mL) was collected from an antecubital forearm vein into a vacutainer tube (EDTA, Becton Dickinson, Oxford, UK). The EDTA blood was centrifuged for 10 min at 1500× *g* and 4 °C, and plasma was stored at −20 °C prior to analysis. After 24 h of blood collection, the acute maximal exercise was performed using an incremental graded cycling exercise test until exhaustion (UB5 Upright Bike, Greenmaster Industrial Corp., Taichung, Taiwan). The pPWV and blood pressure were obtained at rest and 0, 15, 30, 45, 60 min after maximal exercise.

### 2.3. Acute Maximal Exercise

The maximal oxygen consumption ($\dot{V}$O_2max_) was measured using a cortex Gas Analysis System (Metalyzer 3B, Cortex Biophysik GmbH, Sachsen, Germany) during an incremental graded cycling exercise test until exhaustion. Prior to the maximal exercise, the participants were required to perform a 2 min warm-up with no resistance. The incremental graded exercise test was increased by 30 W every 2 min until test termination, starting workload was set as the initial load at 50 W, at a frequency above 60 rpm [22]. The $\dot{V}$O_2max_ (ml/kg/min) obtained for each subject depended on the ability of the participants to achieve at least two of the following criteria: Respiratory exchange ratio (RER) values above 1.15; achieved age-predicted maximum heart rate (208 − 0.7 × age); the pedal frequency cannot maintain 60 rpm; the Borg scale rating of perceived exertion (RPE) is over 17 [23].

### 2.4. Blood Pressure (BP)

Brachial BP was measured at baseline, post-0 min, post-15 min, post-30, post-45 min, and post-60 min using standard sphygmomanometer technique (MEN-1000; Omron Healthy Corp., Taipei, Taiwan). After 5 min of rest time, measurements were taken by a trained researcher on the right arm in the sitting position and repeated after 2 min. The average of the two BP measurements was calculated for statistical analysis [24]. Moreover, a common method used to estimate the mean arterial pressure (MAP) is the following formula: Systolic blood pressure (SBP)—diastolic blood pressure (DBP)/3 + DBP [25].

### 2.5. pPWV

The pPWV was determined by detecting pulse wave by photo-plethysmograph (MP150, Biopac Systems Inc., Goleta, CA, USA), simultaneously with the electrocardiography (ECG) and successively at the right finger and right toe [26]. Two infrared sensors were placed on the right index finger and the second right toe, respectively. Each measurement recorded the digital volume pulses at two locations simultaneously for 150 s. The transit time from finger-to-toe was calculated as the average during this 150 s period. The two measured distances are the difference in the distance measured from the sternal notch to the second right toe and from the sternal notch to the right index finger. The software automatically calculated pPWV (in meters per second) after the finger-to-toe distance measurement was recorded [26].

### 2.6. Plasma Analysis

The best measurement of vitamin D status was considered to be the sum of the 25-hydroxy metabolites of D_2_ and D_3_ (25(OH)D_3_ and 25(OH)D_2_). The 25(OH)D is the main circulating form of vitamin D in the human body, with a circulating half-life of 2 to 3 weeks [3]. The circulating half-life of 1,25(OH)_2_D_3_ and other vitamin-D-related metabolites is shorter than 25(OH)D_3_ and 25(OH)D_2_ at 1000 times [3]. Therefore, the other vitamin-D-related metabolites were not analyzed in this study. The EDTA plasma samples in the study were analyzed for a total 25(OH)D concentrations (25(OH)D_3_ and 25(OH)D_2_). The resting plasma samples were assayed in duplicate in a concentration of 25(OH)D using a commercially available 25-hydroxy Vitamin D^S^ enzyme immunoassay (EIA) (Immunodiagnostic Systems Ltd., Bolton, UK) [27], which was qualified by the Centers for Disease Control and Prevention (CDC) 2019 vitamin D standardization certification [28]. According to the manufacturer’s instructions, 25-hydroxy Vitamin D^S^ EIA is shown to have an excellent agreement with the ‘gold-standard’ high-pressure liquid chromatography–tandem mass spectrometer method. Passing–Bablok regression analyses were reported for the assay from 25-hydroxy Vitamin D^S^ EIA with a mean bias <3% compared to LC-MS/MS. The intra-assay CV was <5% across a working range from 0 to 260 nmol/L [29].

### 2.7. Statistical Analysis

Shapiro–Wilk test was used to analyze the distribution of data sets. The independent-samples t-test was applied to compare the differences in anthropometric, self-reported training loads, physiological variables, and total 25(OH)D concentrations between groups. The pPWV and blood pressure variables were analyzed with two-way analysis (ANOVA). Significant interaction and main effects were followed up with post hoc comparisons between the deficient group and sufficient group using paired-samples t-test for within-group and independent-samples t-test for between-group comparisons. Pearson product–moment analysis was used to observe bivariate correlations between 25(OH)D levels and pPWV. Data are presented as mean ± standard deviation (SD) with statistical analysis completed using SPSS (version 22.0, IBM Inc., Armonk, NY, USA). The accepted level of significance was *p* < 0.05.

## 3. Results

### 3.1. Plasma Total 25(OH)D Concentrations and Baseline Characteristics

The total 25(OH)D concentration from 28 participants classified in the sufficiency group (≥50 nmol/L) was 60.21 ± 6.75 nmol/L and the total 25(OH)D concentration from 22 participants classified in the deficiency group (<50 nmol/L) was 40.68 ± 7.50 nmol/L. There were no significant differences in the $\dot{V}$O_2max_ levels and weekly exercise volume at baseline between the sufficiency and deficiency vitamin D groups (Table 1).

### 3.2. pPWV Responses Following Acute Maximal Exercise

The results show significant effects of time (*p* = 0.000) and groups (*p* = 0.002) on pPWV responses, while there was no significant interaction effect (*p* = 0.177). The t-test analysis revealed that 25(OH)D deficiency group had higher pPWV at post-exercise (5.34 ± 0.71 vs. 4.79 ± 0.81 m/s, *p* < 0.05), post-exercise 15 min (5.13 ± 0.53 vs. 4.48 ± 0.66 m/s, *p* < 0.05) and post-exercise 30 min (5.26 ± 0.84 vs. 4.78 ± 0.50 m/s, *p* < 0.05) than the sufficiency group (Figure 1). In addition, the pPWV significantly decreased from baseline to post-exercise (deficient group: 4%, sufficient group: 8%), post-exercise 15 minute (deficient group: 7%, sufficient group: 14%) and post-exercise 30 minute (deficient group: 6%, sufficient group: 8%) in both groups.

### 3.3. Correlations between Vitamin D Levels and pPWV

The 25(OH)D levels were significantly associated with pPWV at post-exercise (r = −0.364, *p* = 0.011), post-exercise 15 min (r = −0.466, *p* = 0.011) and post-exercise 30 min (r = −0.329, *p* = 0.022). The other levels were not significantly correlated with pPWV at rest (r = −0.217, *p* = 0.139), post-exercise 45 min (r = −0.202, *p* = 0.169) and post-exercise 60 min (r = −0.271, *p* = 0.062) (Figure 2).

### 3.4. Blood Pressure Following Acute Maximal Exercise

The results show the significant effect of time on SBP and MAP, while there was no significant interaction effect and group effect. SBP and MAP significantly increased from pre-exercise after acute maximal exercise, and reached the highest level at post-exercise in both groups. Thereafter SBP and MAP had a declining trend in both groups. DBP had no significant time effect (*p* = 0.437) and group effect (*p* = 0.693) (Table 2).

## 4. Discussion

This is the first study to find that healthy young men with 25(OH)D deficiency had a higher pPWV at post-exercise, post-exercise 15 min, and post-exercise 30 min, than 25(OH)D sufficiency. Additionally, there were significant negative correlations between 25(OH)D levels and pPWV at post-exercise, post-exercise 15 min, and post-exercise 30 min. The results indicate that even healthy young men who exercise regularly still have a poor vascular function when vitamin D is deficient.

The results of our study are similar to previous studies that showed a decrease in pPWV after acute aerobic exercise compared to pre-exercise [22,30]. The effects of acute aerobic exercise on pPWV have been extensively investigated. Acute maximal exercise is known to cause significant changes in arterial stiffness and hemodynamic and may reveal vascular abnormalities not seen at rest [12,13]. Additionally, high-intensity exercise is demonstrated to diminish the magnitude of wave reflections in healthy normal-weight individuals, which means that exercise has the effect of reducing arterial stiffness, ventricular dynamics, and peripheral vasodilation [31]. Thus, the evaluation of arterial stiffness in response to acute exercise could be used as a functional tool for the detection of subclinical vascular dysfunction. In our study, it was found that, following acute maximal exercise, adults with 25(OH)D sufficiency could have a better vasodilation of peripheral arteries (post-15 min pPWV in deficient group 5.13 ± 0.53 m/s versus sufficient group 4.48 ± 0.66 m/s). This may be due to greater vasodilation and less resistance in the peripheral vascular. Mitchell et al. [32] showed that greater vasodilation would reduce the augmentation index (AIx) by the overlap between forward and reflected waves in the central artery (AIx is an indicator of arterial stiffness via wave reflection). Our data suggest that healthy adults with 25(OH)D sufficiency may have increased peripheral vasodilation following acute maximal exercise compared with 25(OH)D deficiency, which may further influence the central arterial stiffness response [22].

Plasma 25(OH)D deficiency or insufficiency was found to be a substantially prevalent condition disorder in the general population [1,2]. It is also known that endothelial and smooth muscle cell function, mediates inflammation are influenced by vitamin D, and modulates the renin–angiotensin–aldosterone axis [33]. In our study, 25(OH)D levels were significantly and negatively correlated with pPWV at post-0, post-15 and post-30 min. Vitamin D receptors and 1-alpha-hydroxylase converts vitamin D into the hormone 1,25-dihydroxy vitamin D form. This is present in many tissues, including vascular endothelial cells [34]. The renin–angiotensin–aldosterone axis has an anti-proliferative effect and is regulated by vitamin D, which influences the vascular wall and affects vascular smooth muscle [35]. In terms of harmful effects, vitamin D deficiency is associated with higher pulse pressure. The individuals with vitamin D deficiency had a poor arterial compliance [36]. In addition, Al Mheid et al. [33] proved that increased arterial stiffness and endothelial dysfunction are related to low 25(OH)D (decrease in reactive hyperemia index (RHI) and flow-mediated dilation (FMD)). Vitamin-D-deficient status might precipitate an individual’s vascular dysfunction and a higher risk for the development of cardiovascular disease and adverse events.

Post-exercise pPWV is likely to reflect the stiffness in the fully dilated peripheral vessels due to increased shear stress exercise-induced hyperemia. Vascular dysfunction caused lower pPWV by increasing vessel diameter, decreasing vasomotor tone, and declining vessel wall thickness [37]. Trachsel et al. [20] found that the response in pPWV to acute maximum exercise seen in healthy young adults was absent in coronary artery disease (CAD) patients. They demonstrate that pPWV did not decrease after acute maximum exercise in the CAD group; however, it significantly decreased in the healthy men group. There was a similar finding in our study that the vitamin D sufficiency group had a greater reduction in pPWV than the deficiency group. The pPWV in the sufficient group decreased by 14% at post-15 minute exercise, while the deficiency group only had 7% decrease. Therefore, the degree of decline in pPWV after maximum exercise could be regarded as a warning of endothelial dysfunction [18]. Even healthy men who exercised regularly showed poorer vascular function when vitamin D was deficiency, which indicates the importance of vitamin D for cardiovascular function [33].

## 5. Limitations

Several limitations of this study deserve to be mentioned. We used a sitting position to measure blood pressure and a lying down position to measure pPWV due to equipment limitations. We tried to curtail the distance between the two instruments as much as possible [38]. We did not include female participants. As we did not measure endothelial function, the level of inflammation and angiotensin level at pre-exercise and post-exercise may support a physiological mechanism for our findings. Further investigation is needed to demonstrate this possible mechanism.

## 6. Conclusions

In this study, we suggested that adults with sufficient 25(OH)D could have better vasodilation of the peripheral arteries after acute aerobic exercise. Vitamin D may play an important role in cardiovascular function that influences endothelial and smooth muscle cell function. The degree of decline in pPWV after acute aerobic exercise could be regarded as a warning of endothelial dysfunction. Vitamin D deficiency may increase the risk of incident cardiovascular events after acute exhaustive exercise, even in healthy and active adults.

## Figures and Tables

**Figure 1 nutrients-14-03074-f001:**
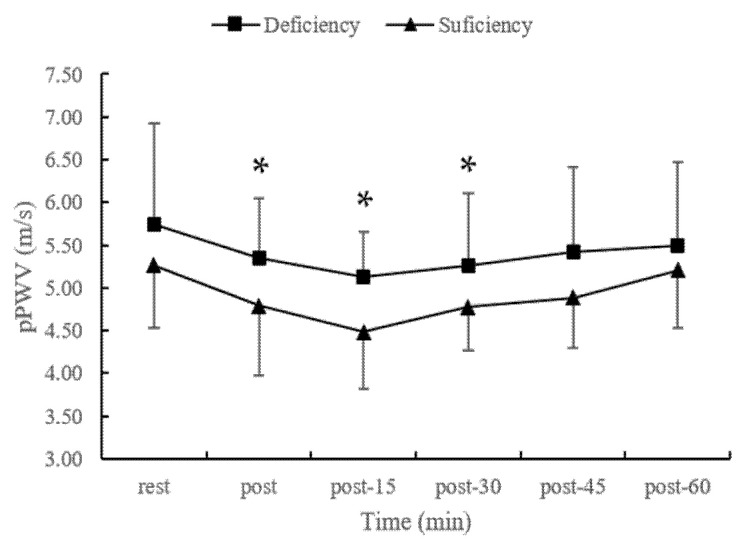
The changes in pPWV following acute maximal exercise in different vitamin D status. Abbreviations: pPWV, peripheral pulse wave velocity. * *p* < 0.05 Significant differences between sufficiency and deficiency group.

**Figure 2 nutrients-14-03074-f002:**
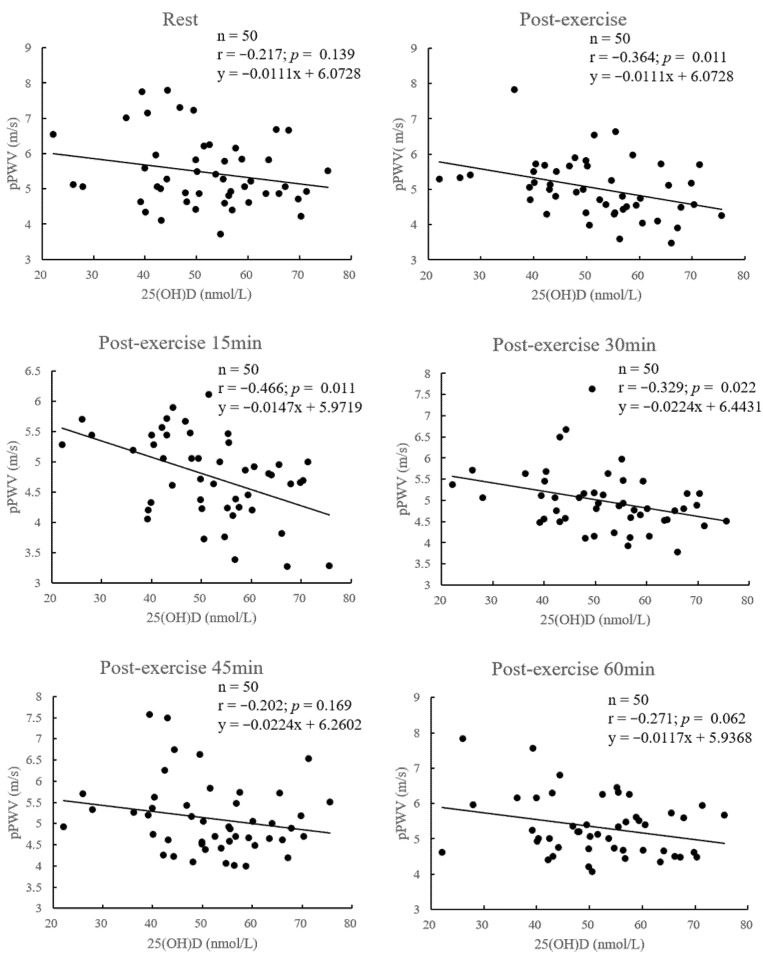
Relationship between vitamin D status and pPWV following exercise at rest and 0, 15, 30, 45, 60 min. Abbreviations: pPWV, peripheral pulse wave velocity.

**Table 1 nutrients-14-03074-t001:** Participant characteristics.

Variable	Sufficiency	Deficiency
Age (years old)	20.96 ± 1.78	21.36 ± 2.12
BMI (kg/m^2^)	23.17 ± 2.61	22.97 ± 2.92
Rest-pPWV (m/s)	5.26 ± 0.72	5.75 ±1.18
Vitamin D (nmol/L)	60.21 ± 6.75 *	40.68 ± 7.5
$\dot{V}$O_2max_ (mL/kg/min)	43.97 ± 5.69	40.78 ± 7.17
Activity (minute/week)	534.29 ± 271.35	519.55 ± 300.75

Abbreviations: BMI, body mass index; pPWV, peripheral pulse wave velocity; $\dot{V}$
O_2max_, maximal oxygen uptake. * *p* < 0.05 Significantly different from deficiency.

**Table 2 nutrients-14-03074-t002:** The changes in blood pressure following acute maximal exercise in different vitamin D status.

		Rest	Post-0 min	Post-15 min	Post-30 min	Post-45 min	Post-60 min	Trial; Time; Trial × Time
SBP (bpm)	Deficient group	121.47 ± 10.70	150.05 ± 13.98 *	119.26 ± 17.17	115.47 ± 9.85 *	112.63 ± 13.37 *	113.58 ± 13.52 *	0.204; 0.000; 0.177
	Sufficient group	126.96 ± 9.32	158.38 ± 15.95 *	123.88 ± 10.76	115.75 ± 12.16 *	111.92 ± 13.32 *	119.38 ± 10.42 *	
DBP (bpm)	Deficient group	63.84 ± 9.05	60.37 ± 11.48	60.58 ± 9.61	61.32 ± 7.55	62.00 ± 10.02	61.32 ± 9.66	0.693; 0.437; 0.575
	Sufficient group	64.08 ± 8.60	63.38 ± 10.84	62.79 ± 10.13	60.79 ± 10.13	59.83 ± 8.71	63.46 ± 6.66	
MAP (bpm)	Deficient group	83.05 ± 8.75	90.26 ± 9.60 *	80.14 ± 10.91	79.37 ± 7.38	78.88 ± 8.28 *	78.74 ± 9.25 *	0.389; 0.000; 0.199
	Sufficient group	85.04 ± 7.52	95.04 ± 10.35 *	83.15 ± 9.06	79.11 ± 7.75 *	77.19 ± 9.17 *	82.10 ± 6.77	

Note: Values are given as mean ± SD. Abbreviations: bpm, beats per minute; DBP, diastolic blood pressure; MAP, mean arterial pressure; SBP, systolic blood pressure. * *p* < 0.05 Significantly different from baseline.

## Data Availability

Data supporting the reported results are available upon reasonable request from C.-S.H.

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
