# Peer review of "Low Vitamin D Status Relates to the Poor Response of Peripheral Pulse Wave Velocity Following Acute Maximal Exercise in Healthy Young Men"

_nutrients, 2022, doi:10.3390/nu14153074_

Round 1
Reviewer 1 Report
Authors in their study try to demonstrate how vitamin D status relates to the poor response of pPWV following maximal exercise in healthy young men. At the begining I would like to say that scope of this work is quite interesting but there are lots of limitations that affects obtained results.
First one:
Authors used two populations, one classified as a deficiency group (25(OH)D < 50 nmol/L), second one as sufficiency group (25(OH)D ≧ 50 nmol/L). Why such ranges of values ​​were used ? It would seem appropriate to actually show a population that suffered from vitamin D deficiency (< 20 nmol/L). Most experts agree that 25(OH)D of < 20 ng/ml is considered to be vitamin D deficiency whereas a 25(OH)D of 21-29 ng/ml is considered to be insufficient. Concentrations of 40 nmol/L are considered to be normal. In the physical culture reserche, the problem of optimizing the concentration of vitamin D is often discussed, but values ​​above 50, 60 nmol/L are considered to be those that may affect the level of physical fitness. So mayby this is the main reson why authors obtained souch results.
Second one: If we disscus about vitamin D metabolis we should messure not only one metabolit, but those that may affect physical health ex : 25(OH)D3, 24,25(OH)2D3, 3-epi-25(OH)D3, and 25(OH)D2 levels and the ratios of 25(OH)D3 to 24,25(OH)2D3 and 25(OH)D3 to 3-epi-25(OH)D3. Furthermore the concentrations of vitamin D metabolites should be corrected to change of the plasma volume. Did authors made souch corrections?
Third one: Concentration quantitative analysis should be performed using liquid chromatography coupled with tandem mass spectrometry. Results basing only on ELISA are not considereted to be the most accurate.
Morover authors show only the changes of selected vitamin D metabolite. But we have to have in minde that exercises alone induces the release of several myokines and exerkines, and some of these molecules possibly stimulates body system and tissues. Morover, according to recent studie vitamin D influences the expression of over 1000 genes, which is why it seems so important to determine the limiting concentration of its.
Methodology:
1.There is no description of the methodology for the determination of vitamin D metabolites.
2. Please explaine why authors used two diffrent conditions to messure pressure and PWV (it's allways affects results - some validation of this protocol..?)
Author Response
Revision of Manuscript ID: nutrients-1813517
Author responses to Nutrients Reviewers' Comments:
We thank both reviewers for their helpful suggestions and constructive comments. We have amended the detail and highlighted them in red in our revised manuscript. We hope that our responses to issues below point-by-point will improve the manuscript sufficiently for you to recommend publication.
Reviewer 1 Comments to the Author
Authors in their study try to demonstrate how vitamin D status relates to the poor response of pPWV following maximal exercise in healthy young men. At the begining I would like to say that scope of this work is quite interesting but there are lots of limitations that affects obtained results.
First one: Authors used two populations, one classified as a deficiency group (25(OH)D < 50 nmol/L), second one as sufficiency group (25(OH)D ≧ 50 nmol/L). Why such ranges of values were used? It would seem appropriate to actually show a population that suffered from vitamin D deficiency (< 20 nmol/L). Most experts agree that 25(OH)D of < 20 ng/ml is considered to be vitamin D deficiency whereas a 25(OH)D of 21-29 ng/ml is considered to be insufficient. Concentrations of 40 nmol/L are considered to be normal. In the physical culture reserche, the problem of optimizing the concentration of vitamin D is often discussed, but values above 50, 60 nmol/L are considered to be those that may affect the level of physical fitness. So mayby this is the main reson why authors obtained souch results.
AUTHOR RESPONSE:
Thank you for the comment. The Institute of Medicine (IOM) and the Endocrine Society have defined a cutoff level of 50 nmol/L as vitamin D deficiency. They considered bone health and dietary reference intakes and found that 50 nmol/L is the requirement for vitamin D in nearly all normal healthy individuals. In addition, the study has found that individuals with vitamin D levels below 50 nmol/L have an increased risk of CVD. (please see rows 35-38 & 41-43, page 1 of the revised manuscript).
Second one: If we disscus about vitamin D metabolis we should messure not only one metabolit, but those that may affect physical health ex : 25(OH)D3, 24,25(OH)2D3, 3-epi-25(OH)D3, and 25(OH)D2 levels and the ratios of 25(OH)D3 to 24,25(OH)2D3 and 25(OH)D3 to 3-epi-25(OH)D3. Furthermore the concentrations of vitamin D metabolites should be corrected to change of the plasma volume. Did authors made souch corrections?
AUTHOR RESPONSE:
Thank you for the comments. A total of 25(OH)D3 and 25(OH)D2 is the main circulating form of vitamin D in the human body, with a circulating half-life of 2 to 3 weeks. 25(OH)D is the best indicator to monitor for vitamin D status. The circulating half-life of 1,25(OH)2D3 and other vitamin D related metabolites is shorter than 25(OH)D3 and 25(OH)D2 at 1000 times. Therefore, the other vitamin D related metabolites were not analyzed in this study. (please see rows 146-150, page 4 of the revised manuscript).
Third one: (1) Concentration quantitative analysis should be performed using liquid chromatography coupled with tandem mass spectrometry. Results basing only on ELISA are not considereted to be the most accurate.
(2) Morover authors show only the changes of selected vitamin D metabolite. But we have to have in minde that exercises alone induces the release of several myokines and exerkines, and some of these molecules possibly stimulates body system and tissues. (3) Morover, according to recent studie vitamin D influences the expression of over 1000 genes, which is why it seems so important to determine the limiting concentration of its.
AUTHOR RESPONSE:
(1) Thank you for the comment. We used a commercially available 25-hydroxy Vitamin DS EIA (Immunodiagnostic Systems Ltd, Bolton, UK), which was qualified by the Centers for Disease Control and Prevention (CDC) 2019 vitamin D standardization certification. According to the manufacturer’s instructions, 25-hydroxy Vitamin DS EIA has been shown by the manufacturer to have an excellent agreement with the ‘gold-standard’ high pressure liquid chromatography-tandem mass spectrometer method. Passing–Bablok regression analyses have been reported for the assay from 25-hydroxy Vitamin DS EIA with a mean bias <3% as compared to LC-MS/MS. The intra-assay CV was < 5% across a working range of 0 to 260 nmol/L. (please see rows 154-160, page 4 of the revised manuscript).
(2) Thank you for the comments. The participants arrived after a 3-hour fast from 09:00am-12:00pm and a quiet venous blood sample (6 ml) was collected (please see rows 106-107 & 109-111, page 3 of the revised manuscript).
(3) Thank you for the comment. Besides, based on our response to the first question above, we used 50 nmol/L as a cutoff level for evaluation of vascular function (Zhou et al., 2022). (please see rows 35-38 & 41-43, page 1 of the revised manuscript).
Methodology:
- There is no description of the methodology for the determination of vitamin D metabolites.
- Please explaine why authors used two diffrent conditions to messure pressure and PWV (it's allways affects results - some validation of this protocol..?)
AUTHOR RESPONSE:
- Thank you for the comment. Besides, based on our response to the second question above. The circulating half-life of 1,25(OH)2D3 and other vitamin D related metabolites is shorter than 25(OH)D3 and 25(OH)D2 at 1000 times. Therefore, the other vitamin D related metabolites were not analyzed in this study. (please see rows 146-150, page 4 of the revised manuscript).
- Thank you for the comment. Thanks for the comments. In this study, due to equipment limitations, the blood pressure was measured in a sitting position and the pPWV in a lying position. We have tried to curtail the distance between the two instruments as much as possible. In addition, all the participants strictly followed the same measurement protocols so that the difference between the groups may not be affected by the measurement. (please see rows 279-282, page 8 of the revised manuscript).

Reviewer 2 Report
Is an interesting approach to consider vitamin D status and vascular health. Despite are not other vascular parameters correlated vit 25(OH)vD, than pPWV, it would be of interest to present the influence of physiologcal and pathological factors on vascular health and how vitamin D status might influence the progression of atherosclerosis.
The subjects were recruited between October to December. What is the variation of vitamin D status in that period? Are you sure that the subjects deficient in 25(OH)D are not the ones monitored in December? How would you comment on that?
Check expression error: row 238 - and ? I do nor understand the phrase
row 209 - refer to the training level of the participants. were they having same physical training capabilities/ levels
Author Response
Revision of Manuscript ID: nutrients-1813517
Author responses to Nutrients Reviewers' Comments:
We thank both reviewers for their helpful suggestions and constructive comments. We have amended the detail and highlighted them in red in our manuscript. We hope that our responses to issues below point-by-point will improve the manuscript sufficiently for you to recommend publication.
Reviewer: 2 Comments to the Author
Is an interesting approach to consider vitamin D status and vascular health. Despite are not other vascular parameters correlated vit 25(OH)vD, than pPWV, it would be of interest to present the influence of physiologcal and pathological factors on vascular health and how vitamin D status might influence the progression of atherosclerosis.
The subjects were recruited between October to December. What is the variation of vitamin D status in that period? Are you sure that the subjects deficient in 25(OH)D are not the ones monitored in December? How would you comment on that?
AUTHOR RESPONSE:
Thank you for the comments. After 24 hours of vitamin D measurement, the participants performed acute maximal exercise. Therefore, our study reflects the current response of vitamin D and vascular function. Moreover, the sunlight in Southeast Asia during this period would not significantly alter the participants’ vitamin D status. We chose this season for our experiment to reduce the difference in sunlight in humans’ vitamin D levels. (please see rows 84-85, page 2 of the revised manuscript).
Check expression error: row 238 - and ? I do nor understand the phrase
AUTHOR RESPONSE:
Thank you for your comments, we have amended the phrase that you pointed out. (please see rows 251-257, page 7 of the revised manuscript).
row 209 - refer to the training level of the participants. were they having same physical training capabilities/ levels
AUTHOR RESPONSE:
Thank you for the comment. There were no significant differences in the O2max levels and weekly exercise volume between vitamin D deficiency and sufficiency in this study. Hence, the participants could have the same physical training capabilities. (please see rows 178-179, page 4 of the revised manuscript).
